# Revisiting Emergent Correspondence from Transformers for Self-supervised Multi-frame Depth Estimation

## Abstract

Self-supervised multi-frame depth estimation predicts depth by leveraging geometric cues from multiple input frames. Traditional methods construct cost volumes based on epipolar geometry to explicitly integrate the geometric information from these input frames. Although this approach may seem effective, the epipolar-based cost volume has two key limitations: (1) it assumes a static environment, and (2) requires pose information during inference. As a result, this cost volume fails in real-world scenarios where dynamic objects and image noise are often present, and pose information is unavailable. In this paper, we demonstrate that the cross-attention map can function as a full cost volume to address these limitations. Specifically, we find that training the cross-attention layers for image reconstruction enables them to implicitly learn a warping function within the cross-attention, resembling the explicit epipolar warping used in traditional self-supervised depth estimation methods. To this end, we propose the CRoss-Attention map and Feature aggregaTor (CRAFT), which is designed to effectively leverage the matching information of the cross-attention map by aggregating and refining the full cost volume. Additionally, we utilize CRAFT in a hierarchical manner to progressively improve depth prediction results through a coarse-to-fine approach. Thorough evaluations on the KITTI and Cityscapes datasets demonstrate that our approach outperforms traditional methods. In contrast to previous methods that employ epipolar-based cost volumes, which often struggle in regions with dynamic objects and image noise, our method demonstrates robust performance and provides accurate depth predictions in these challenging conditions.

## 1 Introduction

Depth estimation has been essential for a wide range of computer vision tasks, including autonomous navigation (Geiger et al., 2012), augmented reality (Luo et al., 2020), or 3D reconstruction (Newcombe et al., 2011). Extensive research has been conducted on self-supervised depth estimation, as it eliminates the need for laborious depth annotations or specialized sensors. While studies have explored both single-frame (Godard et al., 2019; 2017) and multi-frame (Watson et al., 2021; Bangunharcana et al., 2023; Ruhkamp et al., 2021) methods, multi-frame depth estimation generally outperforms single-frame approaches by leveraging additional geometric cues between frames.

In self-supervised multi-frame depth estimation (Watson et al., 2021; Feng et al., 2022; Bangunharcana et al., 2023), a common method to leverage this geometric cue is to construct a cost volume based on epipolar geometry. We refer to this as an *epipolar-based cost volume*. This volume represents the matching cost between the target feature and the reconstructed target feature based on various depth hypotheses. The reconstructed target feature is obtained by warping the source features to the target using pixel-wise similarity calculations relative to the epipolar line in the source image. While this method provides valuable depth information based on multi-view geometry, it has two key limitations: **1)** Dynamic objects and image noise can cause misalignment in pixel correspondences along the epipolar line, leading to violations of the epipolar constraint (Watson et al., 2021; Feng et al., 2022). Previous studies have mitigated this issue by incorporating monocular cues (Godard et al., 2019), but this approach remains limited as the final depth prediction perfor-

mance heavily relies on the monocular prior. **2)** The construction of an epipolar-based cost volume requires an additional pose network during training and inference.

To address the limitations of epipolar-based cost volumes, a straightforward approach may be to build a cost volume that calculates pixel-wise similarity for all points between the target and source frames, including those that do not lie on the epipolar line. We refer to this as a *full cost volume*. Full cost volumes are commonly used in supervised image matching tasks, as they effectively convey comprehensive geometric information to the model (Cho et al., 2021; Xu et al., 2022). However, training a full cost volume in a self-supervised manner poses significant challenges due to its lack of constraints (Kim et al., 2018; Rocco et al., 2018), compared to epipolar-based cost volumes. This is primarily because repetitive patterns and background clutter can introduce ambiguities in feature similarity calculations, complicating its adaptation for multi-frame depth estimation tasks.

We draw inspiration from the widely used cross-attention mechanism (Carion et al., 2020; Ruhkamp et al., 2021), which retrieves information by calculating the key that closely matches the query. Our findings demonstrate that the processes of obtaining the cross-attention map and the full cost volume exhibit mathematical similarities. Furthermore, we show that training the cross-attention layers for image reconstruction enables them to implicitly learn a warping function within the cross-attention map, similar to the explicit epipolar warping employed in self-supervised depth estimation. As a result, we argue that the cross-attention learned in this manner enables the cross-attention map to operate as a full cost volume. To address the challenges associated with training a full cost volume in a self-supervised manner, we leverage masked image modeling (Gupta et al., 2023; Weinzaepfel et al., 2022; Bachmann et al., 2022) to facilitate the pretraining of the cross-attention layers. We validate this approach by visualizing the cross-attention map, which demonstrates accurate matching, suggesting that it effectively captures sufficient geometric information. Moreover, since the attention layer is learned without explicit constraints, it does not require additional information, such as camera pose, thereby enhancing its robustness in dynamic scenarios.

In this paper, we propose a novel self-supervised multi-frame depth estimation architecture that replaces the epipolar-based cost volume with a cross-attention map, which can be viewed as an asymmetric full cost volume. However, simply incorporating the cross-attention map into the model introduces challenges related to resolving inaccuracies and noise in the warping information present in the cost volume. To resolve these ambiguities and filter out unnecessary information for depth prediction, we propose CRoss-Attention map and Feature aggregaTor (CRAFT), designed to compress and refine the full cost volume through feature aggregation with respect to the target image. Furthermore, to capture finer details and handle low-resolution cost volumes, we arrange the CRAFT modules in a pyramidal structure, refining the cost volumes hierarchically. This approach is similar to other methods that enhance features in a coarse-to-fine manner (Min et al., 2021).

We evaluate the robustness of our model in dynamic scenarios by testing it in environments with dynamic objects and various types of image noise, also across overall scenes using the KITTI (Geiger et al., 2013) and Cityscapes (Cordts et al., 2016) datasets. In constrast to traditional self-supervised multi-frame depth estimation methods that rely on epipolar-based cost volumes, our approach fully exploits multi-frame geometric cues from the full cost volume, resulting in enhanced performance in real-world scenarios, particularly in environments with dynamic objects and image noise.

## 2 RELATED WORK

**Self-supervised monocular depth estimation.**   Previous supervised monocular depth estimation methods (Eigen et al., 2014; Eigen & Fergus, 2015; Fu et al., 2018) have achieved notable performance by training models with ground truth depth maps. However, acquiring such maps requires a labor-intensive annotation process or specialized sensors and cameras, which are often not readily available to many researchers. As a result, self-supervised depth estimation methods have emerged as an attractive alternative, framing the training pipeline as an image reconstruction loss using stereo pairs (Xie et al., 2016; Garg et al., 2016; Godard et al., 2017) or monocular video sequences (Godard et al., 2017). Despite the success of (Godard et al., 2017), challenges remain in effectively handling occlusion, noise, and dynamic objects in self-supervised depth estimation from video sequence. Further studies have attempted to tackle these challenges by developing methods for robust image reconstruction loss (Gordon et al., 2019; Godard et al., 2019), discrete depth representation (Johnston & Carneiro, 2020), network architecture improvements (Guizilini et al., 2020), and handling moving objects (Gordon et al., 2019; Li et al., 2021; Godard et al., 2019).

**Multi-frame depth estimation.** The aforementioned self-supervised monocular depth estimation methods utilize temporal frames to form an image reconstruction loss but do not leverage this additional information during inference. To make use of this temporal information, early multi-frame methodologies have employed recurrent networks (CS Kumar et al., 2018; Patil et al., 2020; Wang et al., 2019; Zhang et al., 2019) or test-time optimization techniques (Chen et al., 2019; Kuznietsov et al., 2021; Luo et al., 2020). Recent studies (Watson et al., 2021; Feng et al., 2022; Guizilini et al., 2022; Wang et al., 2023; Bangunharcana et al., 2023) has employed temporal information by constructing a cost volume through epipolar geometry. Specifically, given pre-defined or adaptively determined depth hypotheses and a relative pose between the target and source, the source features are warped to align with the target according to each depth hypothesis, allowing for the measurement of similarity for each warped feature. This epipolar-based cost volume calculates the depth likelihood of the target, thereby providing the model with coarse depth information. However, these methods assume static scenes, struggle with moving objects, and are susceptible to noise. Another approach utilizes spatial-temporal attention for multi-frame depth estimation (Ruhkamp et al., 2021). While this approach eliminates the need for an epipolar-based cost volume, it requires auxiliary depth prediction for spatial attention and encounters difficulties when matching objects of similar appearance, resulting in ambiguous attention.

**Masked image modeling** Inspired by the self-supervised masked modeling of BERT (Devlin, 2018) in NLP, various approaches have been developed to pretrain Vision Transformers (Dosovitskiy, 2020) using self-supervised techniques, aiming to capture richer context and finer details for better representation learning. ViT (Dosovitskiy, 2020) introduced masked patch prediction to improve representations, while BEiT (Bao et al., 2021) extended this idea by predicting discrete visual tokens. More recently, MIM approaches (Xie et al., 2022; He et al., 2022a; Bachmann et al., 2022), which involve reconstructing masked regions of an image, have demonstrated significant success as a pretraining strategy, achieving promising results in downstream tasks.

Furthermore, several methods (Gupta et al., 2023; Bachmann et al., 2022; Weinzaepfel et al., 2022) utilize cross-attention layers to reconstruct masked targets by incorporating additional information, using image reconstruction as a means of learning from supplementary data. For instance, Multi-MAE (Bachmann et al., 2022) learns cross-modality predictive coding through self- and cross-attention applied to different modality inputs. CroCo (Weinzaepfel et al., 2022; 2023) learns geometry by reconstructing a masked target image using the source from the same scene. Similarly, SiamMAE (Gupta et al., 2023) improves motion understanding in videos by reconstructing one frame using another frame from the same video sequence.

## 3 Preliminaries

We briefly review existing self-supervised multi-frame depth estimation networks in the context of epipolar-based cost volumes (Watson et al., 2021; Bangunharcana et al., 2023). Given two images $I_t, I_s \in \mathbb{R}^{H \times W \times 3}$, let the encoded features be $F_t, F_s \in \mathbb{R}^{h \times w \times c}$. The network aims to predict the depth map $D_t \in \mathbb{R}^{H \times W}$, where the subscripts $t$ and $s$ denote the target and source, respectively. To leverage the geometric cues between the target and source frames, an epipolar-based cost volume is employed as an effective method, which identifies an epipolar line in the source frame for every pixel in the target frame (see Figure 1 (a)) to capture this geometric relationship.

To construct an epipolar-based cost volume, it is necessary to warp the source view's features to the target view. By using the target depth $D_t$, the relative pose between the two frames $T_{t \rightarrow s}$, and the camera intrinsics $K$, it becomes possible to determine the warping function $\omega_{t \rightarrow s}(\cdot)$ that contains the correspondence information between every pixel in the target frame $i$ and its corresponding pixels in the source frame:

$$\omega_{t \rightarrow s}(i) = K T_{t \rightarrow s} D_t(i) K^{-1} i, \tag{1}$$

The warping function $\omega_{t \rightarrow s}(i)$, which corresponds to each depth hypothesis $P = \{p_1, ..., p_{|P|}\}$, is applied to the source feature $F_s$ to reconstruct the target feature for each depth hypothesis $F_{t \rightarrow s}^P \in \mathbb{R}^{h \times w \times c \times |P|}$:

$$F_{t \rightarrow s}^P(i) = [\text{sampler}(F_s; \omega_{t \rightarrow s}(i))]_{i=1}^{|P|}, \tag{2}$$

obtained through the bilinear sampling operation $\text{sampler}(\cdot)$. These reconstructed features allows us to derive the epipolar-based cost volume $C_{\text{epi}} \in \mathbb{R}^{h \times w \times |P|}$ using the following equation:

$$C_{\text{epi}}(i, p) = F_t(i) \cdot F_{t \rightarrow s}^P(i, p), \tag{3}$$

where $i$ and $p$ is index of target pixel and hypothesis depth. This cost volume generates higher similarity values in regions with a greater likelihood of accurate depth estimation when the pose is correctly predicted, thereby providing the model with coarse depth information.

Despite the benefits offered by an epipolar-based cost volume, it has two key limitations (Watson et al., 2021): **1)** This volume operates under the assumption of a static scene, meaning that dynamic objects and frame noises can lead to misalignments in pixel correspondences along the epipolar line, ultimately resulting in violations of the epipolar constraint. **2)** The construction of the cost volume requires an estimated pose between frames, which must also be available during inference. To address the first limitation, ambiguity regions arising from moving objects, lens flare, and textureless areas are refined using a monocular depth network. However, the effectiveness of this approach in these challenging regions is largely dependent on the accuracy of the monocular depth estimation network. Regarding the second limitation, a pose network is employed during inference to construct the cost volume, making its robustness highly dependent on the accuracy of the estimated pose. Additionally, incorporating the pose network during inference increases computational complexity.

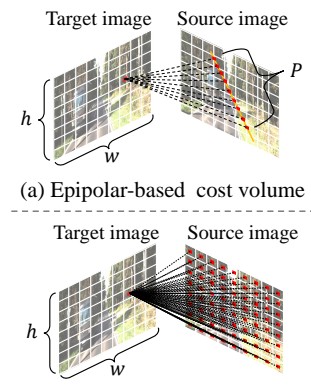

(a) Epipolar-based cost volume

(b) Full cost volume

Figure 1: **Comparison of an epipolar-based cost volume and a full cost volume.**

The depth network $\mathcal{T}_{\text{depth}}$ predicts the depth $D_t = \mathcal{T}_{\text{depth}}(I_t, C_{\text{epi}})$ using the epipolar-based cost volume. In order to train depth estimation in a self-supervised manner, the model leverages correspondences between two frames derived from epipolar geometry. By using the predicted depth and applying a warping function $\omega_{t \to s}(i)$, we can obtain the reconstructed target frame $\hat{I}_{t \to s}$:

$$\hat{I}_{t \to s}(i) = \text{sampler}(I_s; \omega_{t \to s}(i)), \tag{4}$$

where $\text{sampler}(\cdot)$ refers to a bilinear sampling operator. While this reconstruction process is based on the assumption of a static scene, it makes use of explicit geometric information to retrieve an explicit single point from the source frame for reconstructing the target frame. The learning signal is then obtained from the photometric loss $\mathcal{L}_{\text{recon}}$ calculated between the reconstructed target frame and the ground truth target frame:

$$\mathcal{L}_{\text{recon}} = \text{recon}(\hat{I}_{t \to s}, I_t), \tag{5}$$

where $\text{recon}(\cdot, \cdot)$ denotes reconstruction function, such as L1, L2 or SSIM function.

## 4 METHODOLOGY

In this section, We first introduce full cost volumes as an alternative to epipolar-based cost volumes (Sec. 4.1.1). Next, we discuss the similarities between full cost volumes and cross-attention (Sec. 4.1.2), illustrating how cross-attention can implicitly learn geometric cues through self-supervised methods (Sec. 4.1.3). Finally we present the CRAFT module (Sec. 4.2), to address the ambiguity and complexity associated with the full cost volume, as well as the hierarchical structure of CRAFT module (Sec. 4.3) to build a finer cost volume using a hierarchical structure. Our overall architecture is depicted in Figure 2.

### 4.1 CROSS-ATTENTION MAP AS FULL COST VOLUME

#### 4.1.1 FULL COST VOLUME

Epipolar-based cost volumes effectively capture geometric cues by applying constraints that reduce the number of pixel pairs requiring comparison. In contrast, some studies (Cho et al., 2021; Xu et al., 2022) bypass these constraints and instead construct a full cost volume $C_{\text{full}} \in \mathbb{R}^{h \times w \times h \times w}$ by measuring the similarity between all possible pixel pairs, as illustrated in Figure 1 (b):

$$C_{\text{full}}(i, j) = F_t(i) \cdot F_s(j), \tag{6}$$

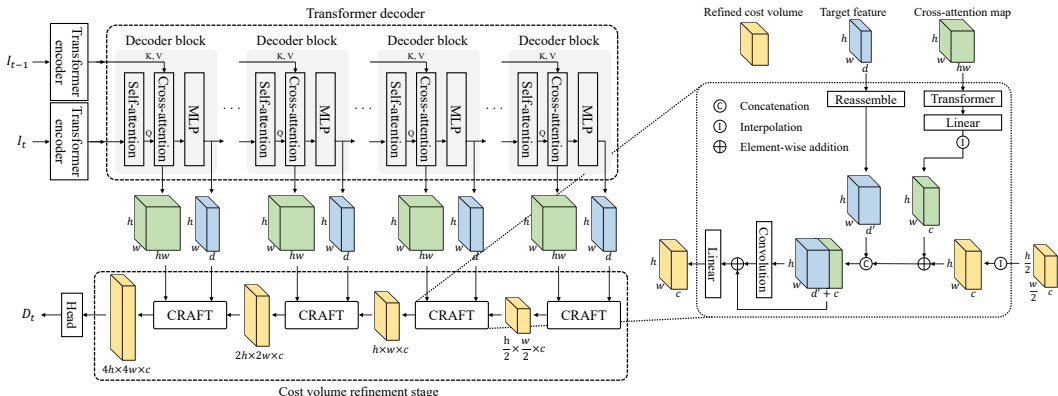

Figure 2: **Overall network architecture.** Our CRoss-Attention map and Feature aggregaTor (CRAFT) module combines target features with the cross-attention map to generate a refined cost volume for depth estimation. To achieve a more detailed depth map, we stack CRAFT module hierarchically which utilizes a four-stage pyramidal structure to address the low resolution of cross-attention maps.

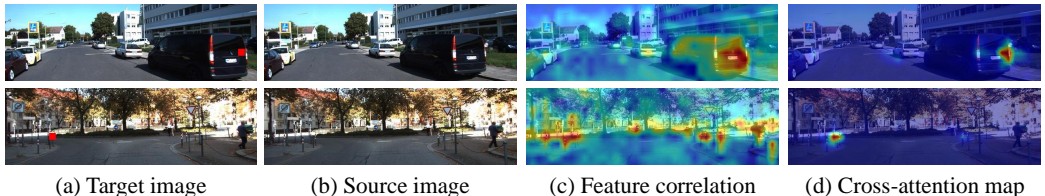

| (a) Target image | (b) Source image | (c) Feature correlation | (d) Cross-attention map |

Figure 3: **Visualization of feature correlation maps and cross-attention maps.** We visualize the matching result of the red dot in the (a) target image in both the (c) feature correlation map and the (d) cross-attention map. The cross-attention map demonstrates superior matching performance compared to the feature correlation map, thanks to its cross-attention mechanism.

where $i$ and $j$ index the target and source frame. The full cost volume effectively manages dynamic scenes better than epipolar-based cost volumes without relying on any constraints. As this volume is constructed using the dot product of features from a shared encoder, it exhibits a symmetric property, enabling the capture of geometric relationships between source and target frames in both directions. Full cost volumes are frequently employed in supervised image matching tasks, as they effectively provide the model with comprehensive geometric information (Cho et al., 2021; Xu et al., 2022). However, training a full cost volume in a self-supervised manner presents significant challenges due to the absence of constraints compared to epipolar-based cost volumes (Kim et al., 2018; Rocco et al., 2018). This challenge primarily arises from regions with repetitive patterns or background clutter, which complicate accurate feature similarity calculations. Such factors further hinder the adaptation of the full cost volume for multi-frame depth estimation tasks, where precise geometric alignment is critical.

### 4.1.2 REVISITING CROSS-ATTENTION

The cross-attention map obtained by processing the encoded target and source features through the cross-attention layer exhibits a strong similarity to a full cost volume. Cross-attention has mainly been employed in tasks involving multi-modal inputs or interactions between distinct inputs, where it extracts relevant information from key features corresponding to the query features (Kirillov et al., 2023; Carion et al., 2020; Guizilini et al., 2022). Specifically, the similarity information is stored within the cross-attention map, and by examining the equation for the cross-attention map $C_{\text{attn}} \in \mathbb{R}^{h \times w \times h \times w}$, we can observe that it closely resembles a full cost volume:

$$C_{\text{attn}}(i, j) = \text{softmax}(F_t^Q(i) \cdot F_s^K(j)), \tag{7}$$

where $F_t^Q = \mathcal{P}_Q(F_t)$ and $F_s^K = \mathcal{P}_K(F_s)$ denote the projected features obtained using $\mathcal{P}_Q$ and $\mathcal{P}_K$, representing the query and key projections, respectively. $\text{softmax}(\cdot)$ denotes a softmax function.

Both the cross-attention map (Equation 7) and the full cost volume (Equation 6) measure pixel-by-pixel similarity between target and source features. In contrast to the symmetric full cost volume, the cross-attention mechanism projects the query and key into distinct spaces and applies softmax, introducing directionality that captures only the geometry of the source relative to the target (Kirillov et al., 2023; Carion et al., 2020; Guizilini et al., 2022). Consequently, the cross-attention map can be viewed as an asymmetric variant of the full cost volume. From the viewpoint of the full cost volume, the cross-attention layer functions similarly to feature warping, retrieving the most similar target feature from the source based on the information encoded in the cross-attention map.

### 4.1.3 Cross-attention with self-supervision

We demonstrate that cross-attention learned in a self-supervised manner, when tasked with image reconstruction, enables the cross-attention map to function as a full cost volume. Specifically, we hypothesize that when cross-attention receives both the target and source as inputs and outputs a reconstructed target, it learns an implicit warping process. This occurs because the reconstruction of the target through feature warping of the source, guided by the attention map, resembles the method used in self-supervised depth estimation, where the source feature reconstructs the target feature using epipolar warping. Analyzing the feature warping equation of cross-attention which produces the reconstructed target reveals:

$$\hat{F}_{t \to s} = \sum_{j=0}^{hw} C_{\text{attn}}(\cdot, j) \, \text{sampler}(F_s^V; j),$$ (8)

where $\hat{F}_{t \to s}$ denotes the reconstructed target feature, and $F_s^V = \mathcal{P}_V(F_s)$ indicates the projected key features using $\mathcal{P}_V$. This equation, along with epipolar warping (Equation 4), highlights the similarity in performing warping to retrieve features from the source frame using the sampler for reconstructing the target. However, while epipolar warping retrieves only a single point based on explicit geometric information, feature warping gathers information from all points and combines them through a weighted sum, where the weights sum to one. Interestingly, the concept of retrieving a single point in epipolar warping can be viewed as a special case of feature warping's weighted sum, where one weight is one and all others are zero.

While the warping equations for both methods are identical, epipolar warping retrieves only a single point, whereas cross-attention aggregates information from all available points, thereby offering enhanced flexibility. However, this increased flexibility presents challenges in training the model to generate an accurately reconstructed target feature. Fortunately, established methods for training cross-attention, such as masked image modeling (MIM) (Weinzaepfel et al., 2022; Gupta et al., 2023; Bachmann et al., 2022), differ from those used to train full cost volumes. By following the training approach of MIM, we train the model on a large number of frames by reconstructing masked target frames, enabling the cross-attention to learn geometry:

$$\mathcal{L}_{\text{recon}} = \text{recon}(\mathcal{E}(\tilde{F}_{t \to s}), I_t),$$ (9)

where $\mathcal{E}$ refers to the output head that corresponds to the shape of the target frame. To illustrate that the trained cross-attention map has implicitly learned geometric relationships, we provide the visualization results in Figure 3. In comparison to the full cost volume constructed by stacking pretrained encoder features (Figure 3 (c)), the cross-attention map, which is learned through frame reconstruction loss (Figure 3 (d)) exhibits more accurate matching, indicating a deeper understanding of geometry. Furthermore, the attention map focuses on a single point, which can be interpreted as an emergent effect (Wei et al., 2022) arising from training on a large dataset, suggesting that the model has implicitly learned geometry. Additionally, by observing the entire scene rather than being confined to the epipolar line, it effectively captures dynamic regions, making the cross-attention map well-suited for functioning as an asymmetric full cost volume.

## 4.2 CRAFT module

Self-supervised multi-frame depth estimation models utilize cost volumes to offer direct multi-frame cues. Similarly, we intend for our model to leverage multi-frame cue information from the cross-attention map. Since this cost volume is learned implicitly, it can result in inaccurate warping or

introduce image noise. To address these challenges, we present the CRAFT module, which is designed to refine both the cross-attention map and the target feature to enhance the accuracy of final depth predictions. This module includes components for both attention aggregation and feature aggregation.

For attention aggregation, we aim to refine the inaccurate matches embodied in the cross-attention map and reduce its high degree of freedom. We employ a transformer (Vaswani et al., 2017) to globally identify the mismatches and enhance the cross-attention map. Specifically, we obtain a compressed cross-attention map $C'_{\text{attn}} \in \mathbb{R}^{h \times w \times c}$ from the initial cross-attention map $C_{\text{attn}} \in \mathbb{R}^{h \times w \times hw}$ as follows:

$$C'_{\text{attn}} = \mathcal{T}_{\text{attn}}(C_{\text{attn}}),$$

where $\mathcal{T}_{\text{attn}}(\cdot)$ means the attention aggregation, comprising transformer and projection layers.

To further reduce the ambiguities inherent in the cost volume, we designed a feature aggregation component that utilizes the target feature to interact with the cross-attention map, producing a refined cost volume. We first preprocess the initial target feature $F \in \mathbb{R}^{h \times w \times d}$ and then concatenate it with the compressed cross-attention map. This combined input is then used to obtain the refined cost volume $C_{\text{ref}} \in \mathbb{R}^{h \times w \times hw}$ as follows:

$$C_{\text{ref}} = \mathcal{T}_{\text{feat}}([C'_{\text{attn}}, F]),$$

where $\mathcal{T}_{\text{feat}}(\cdot)$ denotes the feature aggregation component composed of multiple convolutional layers. Further details about the architecture are provided in Figure 2 and in Section B.1 of the appendix.

### 4.3 COST VOLUME REFINEMENT

The attention map exhibits significantly low resolution due to the patchifying stage within the transformer architecture. As a result, it becomes challenging to generate a finely detailed depth map using the low-resolution attention map. To address this issue, we stack the CRAFT modules in a pyramidal structure, allowing for hierarchical refinement of the cost volumes, similar to previous methods that enhance features in a coarse-to-fine manner (Min et al., 2021). As illustrated in Figure 2, we employ a four-stage pyramidal process using CRAFT modules. We select four different levels of target features $\{F_l\}_{l=1}^4$ and attention maps $\{C_{\text{attn},l}\}_{l=1}^4$ from the transformer decoder. For hierarchical processing, we adjusted the resolution of each level's target feature and attention map in a coarse-to-fine manner. To achieve this, we applied the Reassemble operation from DPT (Ranftl et al., 2021) and bilinear interpolation. The interpolated target features and attention maps are then aggregated to obtain a refined cost volume:

$$C_{\text{ref},l} = \mathcal{T}_{\text{feat}}([\mathcal{T}_{\text{attn}}(\text{interp}(C_{\text{attn},l} + V_{l-1})), F'_l]), \quad F'_l = \text{Reassemble}_l(F_l),$$

where $l$ denotes the stage number, $\text{interp}$ refers to bilinear interpolation, $\text{Reassemble}_l$ indicates the Reassemble blocks for stage $l$, and $C_{\text{ref},l}$ represents the refined cost volume. Additional details can be found in Section B.2 and Section B.3 of the appendix.

## 5 EXPERIMENTS

### 5.1 IMPLEMENTATION DETAILS

Our encoder and decoder are based on Vision Transformers (Dosovitskiy, 2020), initialized with CroCo-v2 weights (Weinzaepfel et al., 2023). This initialization serves to constrain and utilize the cross-attention maps within the cross-attention layers, enabling them to function as a full cost volume for self-supervised multi-frame depth estimation, as discussed in Section 4.1.3. A detailed description of implementation details, datasets, and metrics can be found in Section A of the appendix.

### 5.2 EVALUATION ON REAL WORLD SETTINGS

#### 5.2.1 EXPERIMENTAL SETUP AND METRICS

Our approach differs from previous methods (Watson et al., 2021; Feng et al., 2022; Guizilini et al., 2022; Bangunharcana et al., 2023) by eliminating the need for both a pose network and a monocular

| Method | Additional networks | Test frames | AbsRel↓ | SqRel↓ | RMSE↓ | RMSElog↓ | $\delta_1$↑ | $\delta_2$↑ | $\delta_3$↑ |
|---|---|---|---|---|---|---|---|---|---|
| Monodepth2 (Godard et al., 2019) | - | 1 | 0.159 | 1.937 | 6.363 | 0.201 | 0.816 | 0.950 | 0.981 |
| ManyDepth (Watson et al., 2021) | $\mathcal{M}$ | 2 (-1, 0) | 0.169 | 2.175 | 6.634 | 0.218 | 0.789 | 0.921 | 0.969 |
| DynamicDepth[†] (Feng et al., 2022) | $\mathcal{M}, \mathcal{S}$ | 2 (-1, 0) | 0.143 | 1.497 | **4.971** | 0.178 | 0.841 | 0.954 | 0.983 |
| **Ours** | - | 2 (-1, 0) | **0.127** | **1.322** | 5.058 | **0.175** | **0.860** | **0.964** | **0.987** |

Table 1: **Quantitative results of dynamic objects in the Cityscapes dataset.** We compare our model with previous single- and multi-frame depth estimation networks on dynamic objects as defined in DynamicDepth. The best is **bold** and the second is underline. † means our reproduced results from the official repository. $\mathcal{M}$ means monocular depth network (Godard et al., 2019) and $\mathcal{S}$ means segmentation network.

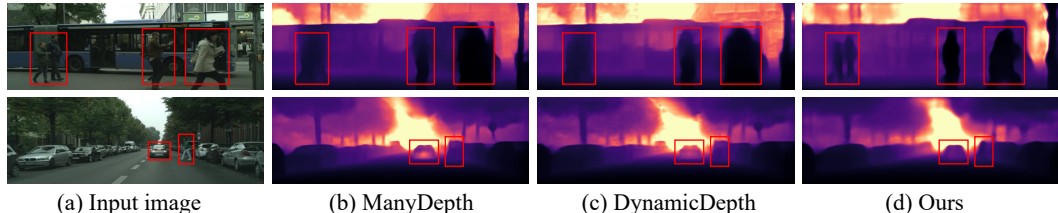

(a) Input image      (b) ManyDepth      (c) DynamicDepth      (d) Ours

Figure 4: **Qualitative results of dynamic objects in the Cityscapes dataset.** In contrast to prior methods that employ epipolar-based cost volumes, our approach leverages full cost volumes to obtain sharper and more accurate depth maps, especially in regions containing dynamic objects, such as moving cars and pedestrians.

prior to construct a multi-frame epipolar-based cost volume. Instead, it allows for the generation of a depth map with a single forward pass through the depth network.

We first conducted evaluations on dynamic objects to demonstrate that our full cost volume provides greater robustness against moving objects compared to epipolar-based cost volumes. Following the evaluation protocol from DynamicDepth (Feng et al., 2022), we measured our model's performance by focusing exclusively on regions within segmentation masks that correspond to moving object labels in the Cityscapes dataset, which features a higher number of dynamic objects. We also evaluated the performance of multi-frame depth estimation under noisy image conditions where epipolar-based cost volumes struggle. Building on the approach of RoboDepth (Kong et al., 2023), we assessed depth performance in practical scenarios with noise, such as motion blur or Gaussian noise. We tested three different scenarios: when noise is present only in the current frame, only in the previous frame, and in both frames. Since it is unlikely that weather changes would affect only a single frame, we focused on noise types identified by RoboDepth, specifically sensor failure and movement, as well as noise introduced during processing. We adopted their evaluation metrics, utilizing mDEE, a combined metric of AbsRel and $\delta_1$, along with mRR, which measures performance degradation compared to noise-free conditions. The detailed equations for mDEE and mRR are described in Section A.4 of the appendix.

### 5.2.2 RESULTS

Table 1 presents a comparison of our model's depth estimation performance on moving objects against other models. While we have addressed the limitations of the epipolar-based cost volume by refining monocular depth estimates, traditional multi-frame depth estimation methods still demonstrate suboptimal performance. Our approach outperforms existing self-supervised multi-frame depth estimation techniques that rely on epipolar-based cost volumes when dealing with moving objects. Notably, our model achieves superior depth estimation results without requiring additional information, in contrast to methods like DynamicDepth and Struct2Depth, which depend on semantic segmentation or optical flow for identifying moving objects. These findings highlight the inherent limitations of epipolar-based cost volumes in dynamic scenarios and underscore the robustness of our full cost volume in such cases. Qualitative results depicted in Figure 4 further illustrate the effectiveness of our approach with dynamic objects. Furthermore, we compared our method against state-of-the-art models, including ManyDepth (Watson et al., 2021) and DualRefine (Bangunharcana et al., 2023). As shown in Table 2, our method demonstrates significantly enhanced resilience to noise. Our experiments revealed that when noise is present solely in the current frame, its impact

| Method | Additional network | Noise frame | mDEE ↓ | mRR ↑ | AbsRel ↓ | SqRel ↓ | RMSE ↓ | $\delta_1$ ↑ |
|---|---|---|---|---|---|---|---|---|
| Manydepth (Watson et al., 2021) | $\mathcal{M}$ | 0 | 0.277 | 0.803 | 0.219 | 1.944 | 7.129 | 0.666 |
| DualRefine (Bangunharcana et al., 2023) | $\mathcal{M}$ | 0 | 0.268 | 0.801 | 0.210 | 1.879 | 7.192 | 0.674 |
| **Ours** | - | 0 | **0.118** | **0.967** | **0.110** | **0.801** | **4.846** | **0.875** |
| Manydepth (Watson et al., 2021) | $\mathcal{M}$ | -1 | 0.118 | 0.979 | 0.113 | 0.885 | 4.667 | 0.878 |
| DualRefine (Bangunharcana et al., 2023) | $\mathcal{M}$ | -1 | 0.102 | 0.983 | 0.101 | 0.750 | **4.421** | 0.897 |
| **Ours** | - | -1 | **0.100** | **0.986** | **0.099** | **0.692** | 4.484 | **0.898** |
| Manydepth (Watson et al., 2021) | $\mathcal{M}$ | 0, -1 | 0.262 | 0.819 | 0.208 | 1.833 | 7.018 | 0.684 |
| DualRefine (Bangunharcana et al., 2023) | $\mathcal{M}$ | 0, -1 | 0.265 | 0.805 | 0.208 | 1.841 | 7.138 | 0.678 |
| **Ours** | - | 0, -1 | **0.161** | **0.920** | **0.137** | **1.043** | **5.556** | **0.816** |

Table 2: **Quantitative results in practical image noises setting on the KITTI dataset.** We followed the evaluation protocol used in Robodepth (Kong et al., 2023) to evaluate the noise robustness. We measured metrics for three different scenarios: when noise is present only in the current frame, only in the previous frame, and in both frames simultaneously. $\mathcal{M}$ means it uses monocular depth network (Godard et al., 2019).

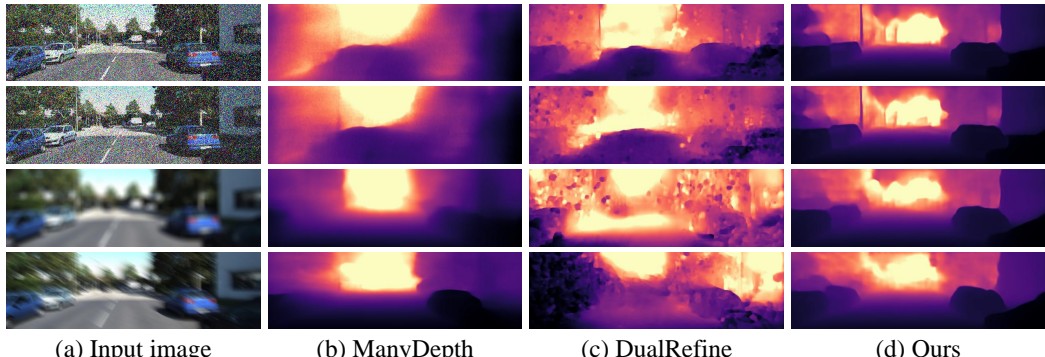

|  (a) Input image  |  (b) ManyDepth  |  (c) DualRefine  |  (d) Ours  |

Figure 5: **Qualitative results in practical image noises setting.** The results are based on applying Gaussian noise, impulse noise, defocus blur, and motion blur to the same image, with each type of noise or blur applied only to the corresponding current frame in the sequence.

on our model's performance is minimal. In contrast, previous multi-frame depth models, which heavily depend on accurate geometric information derived from the epipolar-based cost volume, experience considerable performance degradation in noisy conditions. Our model, however, refines the cost volume through an aggregation process that effectively incorporates contextual information, enhancing its robustness to noise. Figure 5 illustrates the qualitative results, showcasing the stability of our model under various practical noise conditions.

## 5.3 RESULTS ON STANDARD BENCHMARKS

Our primary goal is to demonstrate the robustness of our model in real-world scenarios. However, we also measure our performance on all scenes from the KITTI and Cityscapes datasets to assess its overall depth prediction capabilities. As shown in Table 3 and Table 4, the results indicate that even without using the epipolar-based cost volume, which provides explicit depth information, the full-cost volume can serve as an adequate replacement when utilizing our proposed structure. This strong performance is largely attributed to the robustness of our architecture, particularly in handling moving objects.

## 5.4 ABLATION STUDY

We performed an ablation study on the core components of our model using the KITTI dataset. The results, which evaluate the impact of the attention aggregation module, feature aggregation module, and consistency mask, are presented in Table 5. Additionally, we compared cases where the depth prediction head was our hierarchical CRAFT module structure versus the DPT head (Ranftl et al., 2021), commonly used with transformer-encoded features for depth estimation, trained using static augmentation from ManyDepth (Watson et al., 2021).

| Method | Additional network | Test frames | AbsRel↓ | SqRel↓ | RMSE↓ | RMSElog↓ | $\delta_1\uparrow$ | $\delta_2\uparrow$ | $\delta_3\uparrow$ |
|---|---|---|---|---|---|---|---|---|---|
| Monodepth2 (Godard et al., 2019) | - | 1 | 0.115 | 0.903 | 4.863 | 0.193 | 0.877 | 0.959 | 0.981 |
| Packnet-SFM (Guizilini et al., 2020) | - | 1 | 0.111 | 0.785 | 4.601 | 0.189 | 0.878 | 0.960 | 0.982 |
| MonoViT (Zhao et al., 2022) | - | 1 | 0.099 | 0.708 | 4.372 | 0.175 | 0.900 | 0.967 | 0.984 |
| DualRefine (Bangunharcana et al., 2023) | - | 1 | 0.103 | 0.776 | 4.491 | 0.181 | 0.894 | 0.965 | 0.983 |
| GUDA (Guizilini et al., 2021) | - | 1 | 0.107 | 0.714 | 4.421 | - | 0.883 | - | - |
| RA-Depth (He et al., 2022b) | - | 1 | 0.096 | 0.632 | 4.216 | 0.171 | 0.903 | 0.968 | 0.985 |
| Patil et al. (Patil et al., 2020) | - | N | 0.111 | 0.821 | 4.650 | 0.187 | 0.883 | 0.961 | 0.982 |
| ManyDepth (Watson et al., 2021) | $\mathcal{M}$ | 2 (-1, 0) | 0.098 | 0.770 | 4.459 | 0.176 | 0.900 | 0.965 | 0.983 |
| TC-Depth (Ruhkamp et al., 2021) | $\mathcal{M}$ | 3 (-1, 0, 1) | 0.103 | 0.746 | 4.483 | 0.185 | 0.894 | - | 0.983 |
| DynamicDepth (Feng et al., 2022) | $\mathcal{M}, \mathcal{S}$ | 2 (-1, 0) | 0.096 | 0.720 | 4.458 | 0.175 | 0.897 | 0.964 | **0.984** |
| DepthFormer (Guizilini et al., 2022) | $\mathcal{M}$ | 2 (-1, 0) | 0.090 | 0.661 | 4.149 | 0.175 | 0.905 | 0.967 | **0.984** |
| MOVEDepth (Wang et al., 2023) | $\mathcal{M}$ | 2 (-1, 0) | 0.094 | 0.704 | 4.389 | 0.175 | 0.902 | 0.965 | 0.983 |
| DualRefine (Bangunharcana et al., 2023) | $\mathcal{M}$ | 2 (-1, 0) | **0.087** | 0.698 | 4.234 | 0.170 | **0.914** | 0.967 | 0.983 |
| **Ours** | | 2 (-1, 0) | 0.090 | **0.637** | **4.128** | **0.169** | **0.915** | **0.968** | **0.984** |

Table 3: **Quantitative results on the Eigen split of the KITTI dataset.** We compare our model with previous single- and multi-frame depth estimation networks. For our baseline, we adapted the CroCo-stereo architecture and trained it using a self-supervised depth learning apporach. The best is **bold**, and the second is underline. $\mathcal{M}$ means monocular depth network (Godard et al., 2019) and $\mathcal{S}$ means segmentation network.

| Method | Additional network | Test frames | AbsRel↓ | SqRel↓ | RMSE↓ | RMSElog↓ | $\delta_1\uparrow$ | $\delta_2\uparrow$ | $\delta_3\uparrow$ |
|---|---|---|---|---|---|---|---|---|---|
| Struct2Depth 2 Casser et al. (2019) | - | 1 | 0.145 | 1.737 | 7.280 | 0.205 | 0.813 | 0.942 | 0.976 |
| Monodepth2 Godard et al. (2019) | - | 1 | 0.129 | 1.569 | 6.876 | 0.187 | 0.849 | 0.957 | 0.983 |
| Videos in the Wild Gordon et al. (2019) | - | 1 | 0.127 | 1.330 | 6.960 | 0.195 | 0.830 | 0.947 | 0.981 |
| Li et al. (2021) | - | 1 | 0.119 | 1.290 | 6.980 | 0.190 | 0.846 | 0.952 | 0.982 |
| Struct2Depth 2 Casser et al. (2019) | $\mathcal{M}, \mathcal{F}$ | 3 (-1, 0, 1) | 0.151 | 2.492 | 7.024 | 0.202 | 0.826 | 0.937 | 0.972 |
| ManyDepth Watson et al. (2021) | $\mathcal{M}$ | 2 (-1, 0) | 0.114 | 1.193 | 6.223 | 0.170 | 0.875 | 0.967 | 0.989 |
| DynamicDepth† Feng et al. (2022) | $\mathcal{M}, \mathcal{S}$ | 2 (-1, 0) | **0.104** | **1.009** | **5.991** | **0.150** | **0.889** | 0.972 | **0.991** |
| **Ours** | | 2 (-1, 0) | 0.105 | 1.050 | 6.117 | 0.162 | **0.889** | **0.973** | **0.991** |

Table 4: **Quantitative results on the Cityscapes Dataset.** We compare our model with previous single- and multi-frame depth estimation networks. For our baseline, we adapted the CroCo-stereo architecture and trained it using a self-supervised depth learning apporach. The best is **bold**, and the second is underline. † means our reproduced results from the official repository. $\mathcal{M}$ means monocular depth (Godard et al., 2019), $\mathcal{S}$ means segmentation, and $\mathcal{F}$ means flow network.

| Depth head | CRAFT | | AbsRel↓ | SqRel↓ | RMSE↓ | RMSElog↓ | $\delta_1\uparrow$ | $\delta_2\uparrow$ | $\delta_3\uparrow$ |
|---|---|---|---|---|---|---|---|---|---|
| | $\mathcal{F}_{attn}$ | $\mathcal{F}_{feat}$ | | | | | | | |
| DPT | ✗ | ✗ | 0.102 | 0.767 | 4.351 | 0.766 | 0.903 | 0.960 | 0.978 |
| CRAFT | ✓ | ✗ | 0.150 | 1.235 | 5.515 | 0.221 | 0.808 | 0.935 | 0.975 |
| CRAFT | ✗ | ✓ | 0.092 | 0.650 | 4.218 | 0.178 | 0.910 | 0.963 | 0.981 |
| CRAFT | ✓ | ✓ | **0.090** | **0.637** | **4.128** | **0.169** | **0.915** | **0.968** | **0.984** |

Table 5: **Ablation studies on the KITTI Dataset.**

Our analysis demonstrates that our depth head outperforms DPT head, with significant gains attributed to the integration of the CRAFT module. Upon further examination, we observed that the inherent flexibility of the full cost volume hinders model learning without the attention aggregation module. Furthermore, excluding the feature aggregation module results in degraded performance, as it prevents the refinement of the cost volume necessary for accurate depth estimation.

# 6 CONCLUSION

We have introduced a novel self-supervised multi-frame depth estimation architecture that leverages the CRAFT module to compress and refine the cost volume through attention and feature aggregation. Our results demonstrate that training the cross-attention layers for image reconstruction enables the implicit learning of a warping function within the cross-attention map, similar to the explicit epipolar warping used in previous self-supervised depth estimation methods. Therefore, we conclude that this learned cross-attention enables the cross-attention map to effectively function as a full cost volume. Furthermore, by employing masked image modeling for pretraining, we effectively utilize the cross-attention map as a full cost volume to enhance our depth prediction robustness in dynamic scenarios without the need for additional information, such as camera pose. Evaluations on the KITTI and Cityscapes datasets reveal that our approach outperforms traditional methods using epipolar-based cost volumes, particularly in environments with dynamic objects and image noise.

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

In this appendix, we provide additional implementation details of our framework, along with extended experimental results. In Section A we offers a comprehensive overview of the KITTI and Cityscapes datasets used in our study, as well as a detailed explanation of the metrics for depth estimation and noise settings. In Section B, we elaborates on the design of our Hierarchical Volume Refiner, CRAFT module, and depth head. In Section C, we describe our consistency masking method designed to exclude ambiguous regions from the photometric loss. In Section D, we presents further ablation studies, specifically addressing the impact of freezing encoder or decoder weights and using pretrained weights. Finally, Section E provides expanded qualitative results on the Cityscapes and KITTI datasets, demonstrating that our depth estimation accuracy outperforms other methods, particularly on moving objects.

# A  IMPLEMENTATION DETAILS

## A.1  IMPLEMENTATION DETAILS

We adopt the same augmentation scheme as outlined in (Godard et al., 2019), also using three frames $\{I_{t-1}, I_t, I_{t+1}\}$ for self-supervision during training. Our models are trained with input resolutions of $640 \times 192$ for the KITTI dataset and $416 \times 128$ for the Cityscapes dataset. Experiments were conducted using PyTorch (Aljundi et al., 2019) on an RTX 3090 GPU with a batch size of 8. We employ the Adam (Kingma & Ba, 2014) optimizer, setting the learning rate as $5e-6$ for the pretrained encoder and decoder, and $5e-5$ for the remaining components. We incorporate static augmentation from ManyDepth (Watson et al., 2021) with a $50\%$ ratio. For the photometric loss function, we set $\lambda = 0.001$ and $\alpha = 0.85$.

To leverage the cross-attention map as a full cost volume, we extract both target feature and the cross-attention map from the transformer decoder blocks and feed them hierarchically into our CRAFT module to produce a refined cost volume. When training from scratch, we trained our network using both reprojection loss (Godard et al., 2019) and image reconstruction loss. When performing image reconstruction, we set the mask ratio to 0.5 and applied a weighted sum to the loss by multiplying it by 0.1. Our pose network is implemented as described in (Godard et al., 2019).

## A.2  DATASET

We evaluate our model on KITTI (Geiger et al., 2013) and Cityscapes (Cordts et al., 2016). KITTI is the standard benchmark for depth estimation which is an outdoor dataset of driving scenario. Following (Eigen & Fergus, 2015), we utilize Eigen split with preprocessing from (Zhou et al., 2017) which comprises $39,810$ training data, $4,424$ validation data, and 697 test data. However, 22 frames in test set of Eigen split do not have a previous frame. In such cases, we followed (Watson et al., 2021) by conducting the evaluation using a single frame only.

We also conduct our experiments using the Cityscapes dataset (Cordts et al., 2016). We use a preprocessed set of $69,731$ images for training, as done in (Zhou et al., 2017), and $1,525$ images with disparity maps processed by SGM (Hirschmuller, 2007) for testing. Both of KITTI and Cityscapes dataset, we only evaluate the pixels where the ground truth depth is less than 80 meters and the predicted depth was also clipped to a maximum of 80 meters.

## A.3  METRICS

We use standard depth evaluation metrics from (Eigen & Fergus, 2015; Eigen et al., 2014) with median scaling same as (Zhou et al., 2017). Lower is better for error based metrics (Abs Rel, Sq Rel, RMSE, RMSE log) and higher is better for accuracy based metrics ($\delta_1$, $\delta_2$, $\delta_3$). These are formulated as:

$$\text{AbsRel} = \frac{1}{n} \sum_i \frac{|d_i - \hat{d}_i|}{d_i}.$$

$$\text{SqRel} = \frac{1}{n} \sum_i \frac{(d_i - \hat{d}_i)^2}{d_i},$$

$$\text{RMSE} = \sqrt{\frac{1}{n}\sum_i (d_i - \hat{d}_i)^2}.$$

$$\text{RMSE}_{\log} = \sqrt{\frac{1}{n}\sum_i (\log d_i - \log \hat{d}_i)^2},$$

$$\delta_1, \delta_2, \delta_3 = \max(\frac{d}{\hat{d}}, \frac{\hat{d}}{d}) < 1.25, 1.25^2, 1.25^3$$

where $\hat{d}$ and $d$ means predicted depth and ground truth depth, respectively.

## A.4 Noise Metrics

Building on the methodology of RoboDepth (Kong et al., 2023), which evaluated depth prediction performance in noisy, real-world scenarios, we also assessed the performance of multi-frame depth estimation under challenging conditions where epipolar-based cost volumes tend to struggle. We employed their evaluation metrics, including mDEE, a combined metric of AbsRel and $\delta_1$, as well as mRR, which measures the degradation in performance relative to noise-free conditions. These metrics are defined as follows:

$$\text{DEE} = \frac{\text{AbsRel} - \delta_1 + 1}{2},$$

$$\text{mDEE} = \frac{1}{N \cdot L}\sum_{i=1}^{N}\sum_{l=1}^{L}\text{DEE}_{i,l},$$

$$\text{RR}_i = \frac{\sum_{l=1}^{L}(1 - \text{DEE}_{i,l})}{L \times (1 - \text{DEE}_{\text{clean}})},$$

$$\text{mRR} = \frac{1}{N}\sum_{i=1}^{N}\text{RR}_i,$$

where $L$ denotes the number of severity levels, $N$ denotes the number of augmentations, and $\text{DEE}_{\text{clean}}$ refers to the DEE metric without any augmentation.

To demonstrate the noise robustness of our framework, we evaluated it across five levels of severity using augmentations from the 'Sensor & Movement' and 'Data & Processing' categories. These augmentations included defocus blur, glass blur, motion blur, zoom blur, elastic transformation, color quantization, gaussian noise, impulse noise, shot noise, ISO noise, pixelate, and JPEG compression.

## B Architecture Details

We begin by reviewing the notations and dimensions of the input images, cross-attention map, and target features. Specifically, the resized input images $I_{t-1}, I_t \in \mathbb{R}^{192 \times 640 \times 3}$ are patchified (with a patch size of 16) and then embedded for processing by the transformer. This results in a cross-attention map $A \in \mathbb{R}^{h \times w \times hw}$ and a target feature $F \in \mathbb{R}^{h \times w \times d}$, where $h = 12$, $w = 40$ and $d = 768$. Conventionally, attention maps and features of transformers are expressed in 2D dimensions of $\mathbb{R}^{N \times N}$ and $\mathbb{R}^{N \times d}$, respectively, where $N$ denotes the number of tokens. Here, we express the attention maps and features in 3D dimensions, where $N = hw$, and the channels in the attention maps represent the number of tokens in $I_{t-1}$. Note that the cross-attention map is averaged along the head dimension.

### B.1 CRAFT module

The role of the CRAFT module is to effectively aggregate its inputs and increase their resolution to produce a higher-resolution refined cost volume. The inputs to the CRAFT module include the cross-attention map, the target feature, and, if available, the refined cost volume from the previous CRAFT module. The CRAFT module's output is the refined cost volume. The cross-attention map is compressed using two transformer layers (Dosovitskiy, 2020), while the target feature is compressed

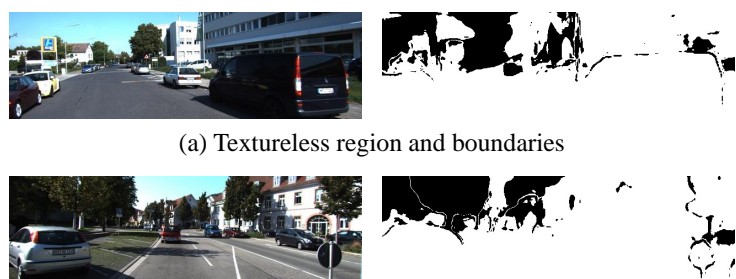

(a) Textureless region and boundaries

(b) Repetitive texture

Figure 6: **Consistency masking visualization.** Our consistency mask (right), corresponding to the input image (left), effectively filters out regions where photometric loss is ambiguous, such as textureless regions, boundaries (a), and repetitive patterns (b).

using the reassemble operation from DPT (Ranftl et al., 2021), which involves convolutions. Bilinear interpolation is used to upsample both the cross-attention map and the input cost volume, while the target feature is upsampled as part of the reassemble operation. This results in attention maps $A' \in \mathbb{R}^{h \times w \times c}$ and target features $F' \in \mathbb{R}^{h \times w \times d'}$, where $d' = 256$. As depicted in Figure 2 of our main paper, these are aggregated by adding them to the previous refined cost volume and then concatenated. This combined cost volume is further processed by two convolution layers, along with residual connections and a projection layer, to produce the refined cost volume.

### B.2 COST VOLUME REFINEMENT STAGE

We leverage cross-attention maps from transformer layers as cost volumes. However, since these maps are derived from patchified images, which are of low resolution, they are unsuitable for pixel-wise depth estimation without modifications. To address this issue, we progressively upsample the refined cost volume to enhance its effectiveness for depth estimation, as demonstrated in previous studies (Ranftl et al., 2021). The final refined cost volume input to the depth head has dimensions of $\mathbb{R}^{4h \times 4w \times c}$, where $c = 128$.

### B.3 DEPTH HEAD

In training, we attach a depth head to each CRAFT module to create a hierarchical depth map that evolves from coarse to fine, ultimately producing a detailed final depth map. Figure 2 in our main paper illustrates only the depth head associated with the final CRAFT module. The depth head comprises two convolution layers, bilinear interpolation, a ReLU activation function, and a sigmoid activation function.

## C CONSISTENCY MASK

We propose a simple yet effective masking method. Typically, we can predict not only multi-frame depth, $D_{\text{multi}} = \theta_D(I_t, I_{t-1})$ but also single-frame depth $D_{\text{static}} = \theta_D(I_t, I_t)$ by providing identical inputs for both frames. Because photometric loss propagates weak supervision, it outputs depth values with high uncertainty in regions such as textureless region (Shu et al., 2020), boundaries (Watson et al., 2019), and repetitive textures (Chen et al., 2023). Therefore, theoretically, the two depth values should be the same, but different values are observed on these regions. So, we filtered these areas during training:

$$\mathcal{M}_c = \left[ \frac{|D_{\text{multi}} - D_{\text{static}}|}{D_{\text{static}}} < k\% \right].$$

where $\mathcal{M}_c$ is our proposed consistency mask and $[< k\%]$ means top-k operation. In Figure 6, we can observe that our consistency mask effectively masks regions such as textureless areas (e.g, sky), boundaries, and repetitive regions(e.g. trees) and it helps the model learn robustly.

| Encoder Freeze | Decoder Freeze | AbsRel↓ | SqRel↓ | RMSE↓ | $\delta_1$↑ |
|:---:|:---:|:---:|:---:|:---:|:---:|
| ✓ | ✓ | 0.112 | 0.821 | 4.611 | 0.881 |
| ✓ | ✗ | 0.097 | 0.678 | 4.324 | 0.904 |
| ✗ | ✗ | **0.090** | **0.649** | **4.201** | **0.915** |

Table 6: **Ablation studies of freezing network on the KITTI Dataset.**

## D  ADDITIONAL ABLATION STUDY RESULTS

### D.1  FREEZING NETWORKS

We performed additional experiments to investigate the impact of freezing the transformer encoder and decoder. Given that our encoder and decoder are well-trained from the cross-view completion task, they embed geometric knowledge and matching information within the attention maps, which we use as full cost volumes. We hypothesize that by utilizing this full cost volume, the model can achieve a certain level of performance.

Table 6 presents the results. The experiments demonstrate that the network effectively utilizes the full cost volume despite the reduction in trainable weights from freezing the encoder and decoder. However, additional training of the encoder and decoder in our architecture allows the model to learn richer full cost volumes, which improves accurate depth estimation accuracy and yields the best results.

### D.2  PRETRAINED WEIGHTS

#### D.2.1  EXPERIMENT SETTINGS

Our model relies on a well-learned full cost volume derived from the cross-view completion training process. Consequently, we utilized the pretrained weights from the cross-view completion model. We present an ablation study comparing the performance of the model trained from scratch with that of the model trained using the CroCo pretrained weights.

Additionally, inspired by the training method used in cross-view completion, we applied a masking technique to the target image during training from scratch. We also incorporated an image reconstruction loss into the transformer decoder to ensure that the attention map in the cross-attention layer implicitly functions as a full cost volume, described as the following:

$$\tilde{F}_t, F_s = \theta_{\text{enc}}(\tilde{I}_t), \theta_{\text{enc}}(I_s)$$

$$\hat{I}_t = \Pi(I_t, I_s; \theta) = \theta_{\text{dec}}(\tilde{F}_t, F_s)$$

where $\theta_{\text{enc}}$ and $\theta_{\text{dec}}$ denote the transformer encoder and decoder, respectively, $\tilde{I}_t$ represents the masked target image, and $\hat{I}_t$ denotes the reconstructed target image. Given $\hat{I}_t$ from the transformer decoder, we compute the masked image reconstruction loss as follows:

$$\mathcal{L}_{img} = ||I_t - \hat{I}_t||,$$

and the final loss for this ablation study is given by:

$$\mathcal{L}' = \mathcal{M} \cdot \mathcal{L}_p + \lambda \mathcal{L}_s + \mu \mathcal{L}_{img},$$

where $\mu$ is the weight for the masked image reconstruction loss.

#### D.2.2  RESULTS

The results of using the pretrained weights are presented in Table 7. As evident from the results, the model trained from scratch exhibited lower performance due to the absence of implicit cost volume information in the cross-attention map. However, by incorporating additional masking and masked image reconstruction loss during training, the attention map from scratch was able to learn partial matching information more effectively, enhancing its ability to function as a cost volume and leading to a noticeable improvement in performance.

| pretrain | mask ratio | loss weight ($\mu$) | AbsRel↓ | SqRel↓ | RMSE↓ | $\delta_1$↑ |
|---|---|---|---|---|---|---|
| Scratch | 0.0 | 0 | 0.120 | 0.872 | 4.899 | 0.866 |
| | 0.25 | 1 | 0.102 | 0.754 | 4.668 | 0.893 |
| | 0.25 | 0.1 | 0.100 | 0.727 | 4.498 | 0.897 |
| | 0.25 | 0.01 | 0.106 | 0.801 | 4.657 | 0.890 |
| | 0.5 | 0.1 | 0.102 | 0.769 | 4.503 | 0.897 |
| | 0.5 | 0.01 | 0.103 | 0.788 | 4.586 | 0.894 |
| CroCo | 0.25 | 0.1 | 0.092 | 0.670 | 4.243 | 0.908 |
| | 0.25 | 0.01 | 0.092 | 0.637 | 4.190 | 0.912 |
| | 0.0 | 0 | **0.090** | **0.637** | **4.128** | **0.915** |

Table 7: **Ablation studies of pretrained weights on the KITTI Dataset.**

| Method | Additional networks | Test frames | AbsRel↓ | SqRel↓ | RMSE↓ | RMSElog↓ | $\delta_1$↑ | $\delta_2$↑ | $\delta_3$↑ |
|---|---|---|---|---|---|---|---|---|---|
| Monodepth2 (Godard et al., 2019) | - | 1 | 0.159 | 1.937 | 6.363 | 0.201 | 0.816 | 0.950 | 0.981 |
| ManyDepth (Watson et al., 2021) | $\mathcal{M}$ | 2 (-1, 0) | 0.169 | 2.175 | 6.634 | 0.218 | 0.789 | 0.921 | 0.969 |
| DynamicDepth[†] (Feng et al., 2022) | $\mathcal{M}, \mathcal{S}$ | 2 (-1, 0) | 0.143 | 1.497 | **4.971** | 0.178 | 0.841 | 0.954 | 0.983 |
| **Ours-Scratch** | - | 2 (-1, 0) | 0.171 | 2.316 | 6.132 | 0.215 | 0.818 | 0.939 | 0.972 |
| **Ours-pretrain** | - | 2 (-1, 0) | **0.127** | **1.322** | 5.058 | **0.175** | **0.860** | **0.964** | **0.987** |

Table 8: **Quantitative results of dynamic objects in the Cityscapes dataset.** We compare our model with previous single- and multi-frame depth estimation networks on dynamic objects as defined in DynamicDepth. The best is **bold** and the second is underline. † means our reproduced results from the official repository. $\mathcal{M}$ means monocular depth network (Godard et al., 2019) and $\mathcal{S}$ means segmentation network.

| Method | Additional network | Noise frame | mDEE ↓ | mRR ↑ | AbsRel ↓ | SqRel ↓ | RMSE ↓ | $\delta_1$ ↑ |
|---|---|---|---|---|---|---|---|---|
| Manydepth (Watson et al., 2021) | $\mathcal{M}$ | 0 | 0.277 | 0.803 | 0.219 | 1.944 | 7.129 | 0.666 |
| DualRefine (Bangunharcana et al., 2023) | $\mathcal{M}$ | 0 | 0.268 | 0.801 | 0.210 | 1.879 | 7.192 | 0.674 |
| **Ours-Scratch** | - | 0 | 0.116 | 0.988 | 0.112 | 0.873 | 4.795 | 0.879 |
| **Ours-Pretrain** | - | 0 | **0.118** | **0.967** | **0.110** | **0.801** | **4.846** | **0.875** |
| Manydepth (Watson et al., 2021) | $\mathcal{M}$ | -1 | 0.118 | 0.979 | 0.113 | 0.885 | 4.667 | 0.878 |
| DualRefine (Bangunharcana et al., 2023) | $\mathcal{M}$ | -1 | 0.102 | 0.983 | 0.101 | 0.750 | **4.421** | 0.897 |
| **Ours-Scratch** | - | -1 | 0.120 | 0.984 | 0.113 | 0.846 | 4.961 | 0.873 |
| **Ours-Pretrain** | - | -1 | **0.100** | **0.986** | **0.099** | **0.692** | 4.484 | **0.898** |
| Manydepth (Watson et al., 2021) | $\mathcal{M}$ | 0, -1 | 0.262 | 0.819 | 0.208 | 1.833 | 7.018 | 0.684 |
| DualRefine (Bangunharcana et al., 2023) | $\mathcal{M}$ | 0, -1 | 0.265 | 0.805 | 0.208 | 1.841 | 7.138 | 0.678 |
| **Ours-Scratch** | - | 0, -1 | 0.151 | 0.949 | 0.133 | 1.030 | 5.441 | 0.830 |
| **Ours-Pretrain** | - | 0, -1 | **0.161** | **0.920** | **0.137** | **1.043** | **5.556** | **0.816** |

Table 9: **Quantitative results in practical image noises setting on the KITTI dataset.** We followed the evaluation protocol used in Robodepth (Kong et al., 2023) to evaluate the noise robustness. We measured metrics for three different scenarios: when noise is present only in the current frame, only in the previous frame, and in both frames simultaneously. $\mathcal{M}$ means it uses monocular depth network (Godard et al., 2019).

On the other hand, applying masking and masked image reconstruction loss to a model initialized with CroCo pretrained weights resulted in a slight decrease in performance. This reduction is likely because the implicit cost volume information already embedded in the cross-attention map of CroCo was distorted by the masking process, which potentially impaired the learned matching information.

## D.3 RESULTS IN PRACTICAL SETTINGS

We also aim to demonstrate the robustness of our model by evaluating not only the version initialized with pretrained weights but also the one trained from scratch. As shown in Table 8, our model exhibits stronger performance on moving objects compared to Manydepth, even without any priors. Furthermore, as seen in Table 9, our training method proves to be more resilient to noise compared to epipolar-based cost volumes. In some cases, it even outperforms the model trained with pretrained weights.

# E   ADDITIONAL QUALITATIVE RESULTS

We present qualitative results for the Cityscapes dataset in Figure 7 and additional qualitative results for the KITTI dataset in Figure 8. For Cityscapes, we compare our method with Manydepth (Watson et al., 2021) and Dynamicdepth (Feng et al., 2022), whereas for KITTI, we compare it with Manydepth (Watson et al., 2021) and Dualrefine (Bangunharcana et al., 2023). Our qualitative results show that our model accurately predicts depth for moving objects, such as pedestrians and cars, while also effectively capturing distant objects.

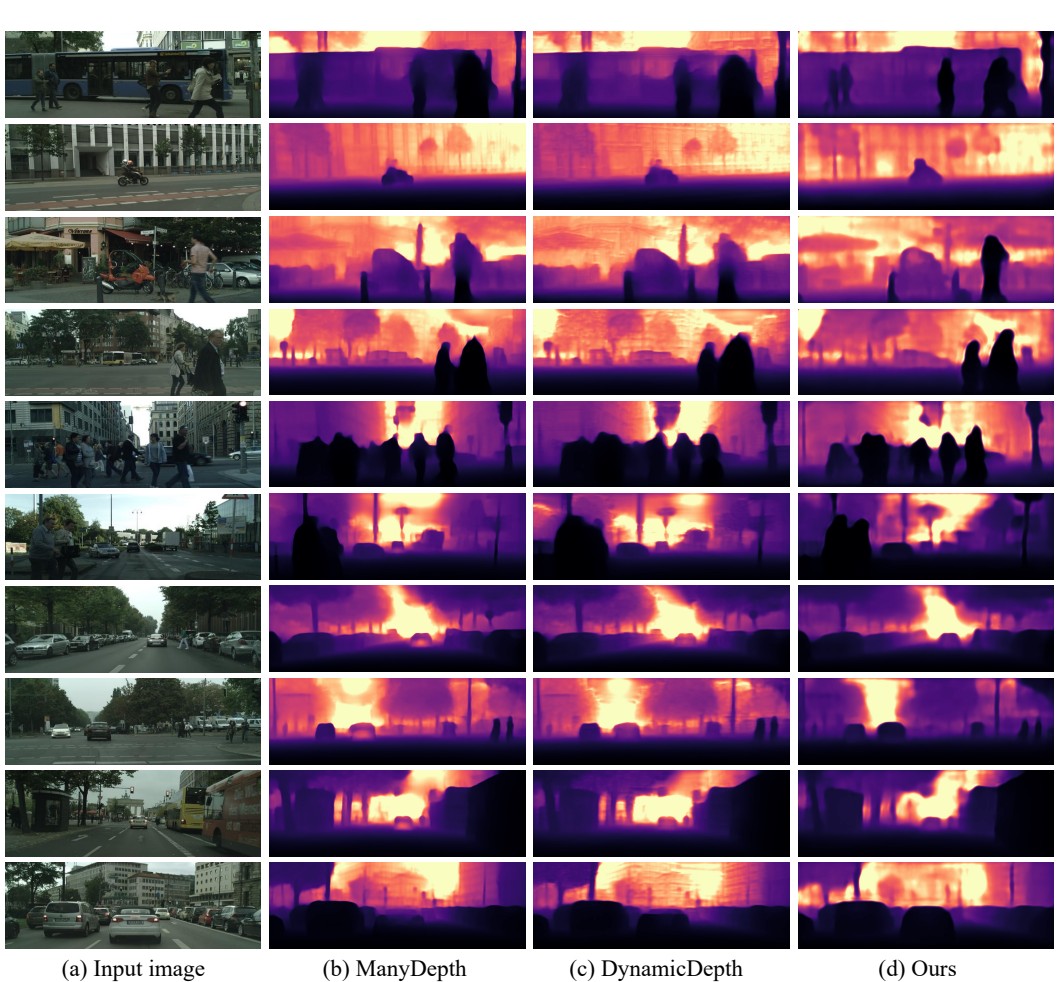

(a) Input image        (b) ManyDepth        (c) DynamicDepth        (d) Ours

Figure 7: **Qualitative results on Cityscapes dataset.**

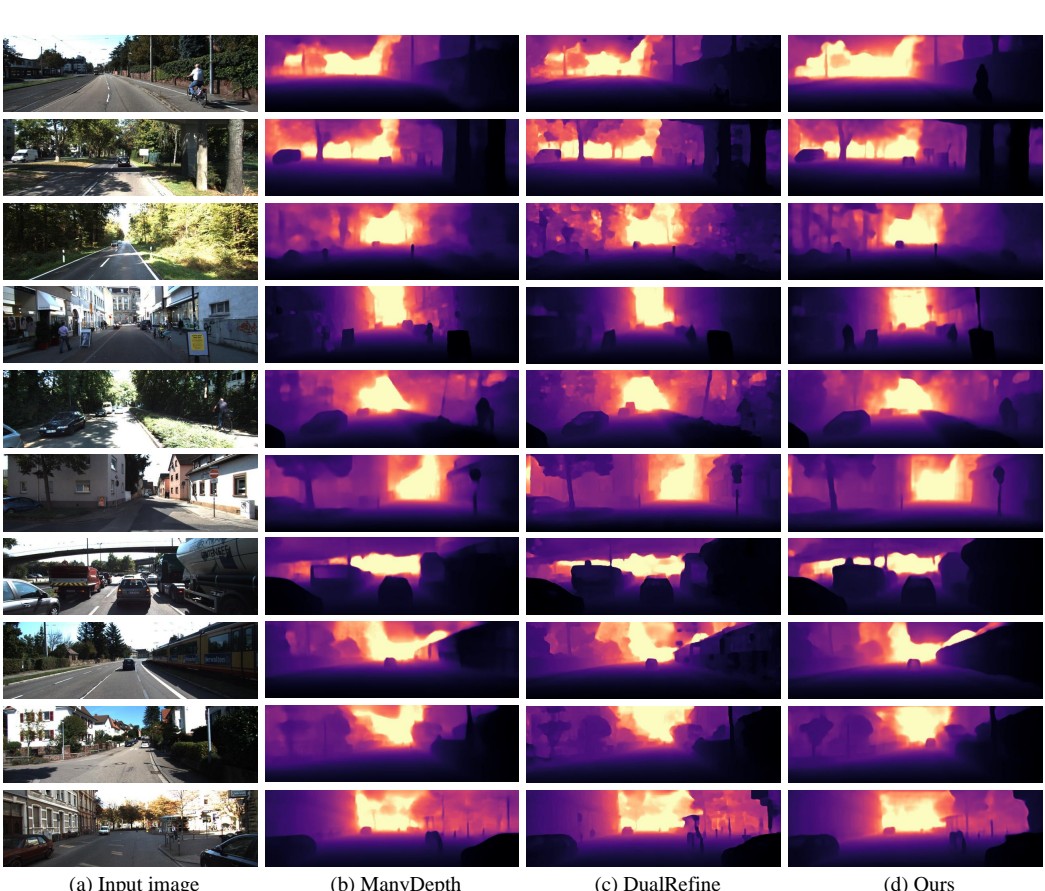

(a) Input image          (b) ManyDepth          (c) DualRefine          (d) Ours

Figure 8: **Additional qualitative results on KITTI dataset.**

