# OpenReview forum: "Revisiting Emergent Correspondence from Transformers for Self-supervised Multi-frame Depth Estimation"
_ICLR.cc/2025/Conference — ICLR 2025 Conference Withdrawn Submission_

### Official Review · Reviewer_1j4C · 2024-10-30

**Soundness:** 2
**Presentation:** 3
**Contribution:** 2
**Rating:** 3
**Confidence:** 5

**Summary:**

This paper proposes a self-supervised multi-frame depth estimation framework, which introduces the cross-attention mechanism to implicitly learn the warping function instead of explicit epipolar warping. To this end, this paper proposes the CossAttention map and Feature aggregaTor (CRAFT), which is designed to effectively leverage the matching information of the cross-attention map by aggregating and refining the full cost volume. In addition, the CRAFT is used in a hierarchical manner to improve the depth prediction. Evaluations on the KITTI and Cityscapes datasets demonstrate that this work effectively in environments with dynamic objects and image noise.

**Strengths:**

1. The idea is simple, clear and easy to reproduce.
2. The writing is excellent.

**Weaknesses:**

1.The method presented in this paper lacks novelty. The cross-attention mechanism or the transformer architecture has been widely used in depth estimation task, and can improve depth quality well. Using a cross-attention mechanism instead of traditional warping and coarse-to-fine strategies is not particularly novel.
2.Performance improvements are limited. The work is based on a transformer architecture, while most of the compared methods are based on CNN architecture such as ResNet and HRNet， which is unfair. Moreover, the quantitative results in Table 4 are worse than the 2022 work and the recent work [1].

[1] Miao X, Bai Y, Duan H, et al. Ds-depth: Dynamic and static depth estimation via a fusion cost volume[J]. IEEE Transactions on Circuits and Systems for Video Technology, 2023.

**Questions:**

1.	Compare more recent methods in Cityscapes.
2.	Please compare fairly. Comparison on fair feature extractors.
3.	Compare model complexity and inference time.
4.	Other questions refer to weakness.

---

### Official Review · Reviewer_pyra · 2024-11-01

**Soundness:** 3
**Presentation:** 3
**Contribution:** 2
**Rating:** 3
**Confidence:** 4

**Summary:**

The authors target at the self-supervised multi-frame monocular depth estimation. Besides, they propose to use the cross-attention to replace conventional cost volume. The proposed method is validated on the KITTI and Cityscapes datasets.

**Strengths:**

The paper is well-written and structured;

The experiments are thorough and conducted on multiple datasets;

The proposed method outperforms previous state-of-the-art methods;

**Weaknesses:**

Using cross-attention to replace the cost volume has been explored in previous methods, such as [1]:

[1] Revisiting Stereo Depth Estimation From a Sequence-to-Sequence Perspective with Transformers, ICCV 2021.

Could you provide a detailed explanation of the structural differences between the proposed method and CRFAT? It currently appears that the differences are mainly at the output level.

Additionally, could you offer a comparison of the computational complexity between the proposed method and ManyDepth?

**Questions:**

Please see the weakness section.

---

### Official Review · Reviewer_R7Vf · 2024-11-01

**Soundness:** 3
**Presentation:** 3
**Contribution:** 2
**Rating:** 5
**Confidence:** 1

**Summary:**

The Paper deals with using Transformer architecture for estimating depth using multiple frames in a self-supervised setting, without using explicit pose information. Paper Proposes to use Full Cost Volume instead of Epipolar Cost Volume. Employs MIM-based pre-training and then employs Cross-Attention. Paper claims to deal with noise much better and does not need an explicit pose network or pose information

**Strengths:**

1. The Paper is well written, outlining the shortcomings of contemporary/previous works and explains clearly the approach followed by the paper to overcome those shortcomings.
2. Proposes to use Full Cost Volume instead of Epipolar Cost Volume. Employs MIM-based pre-training and then employs Cross-Attention.
3. Evaluates the Model performance on relevant datasets to point-out the improvement in the shortcomings of previous methods on KITTI and Cityscapes dataset.

**Weaknesses:**

1. Qualitative results provided in the Figure 4 is 'not 'good'. Where there is improvement in depth prediction where the Figure 4 highlights using the red boxes but on non-highlighted region of the depth prediction is seen, we see worse results.
a. Trees in the second image is definetly not sharper compared to Dynamicdepth & Manydepth as presented in the paper.
b. Buildings in the background  are much more clearer for DynamicDepth and Manydepth for the first image.

2. Since the tasks proposed here is very close to Multi View Stereo tasks, Could the authors provide  the 3D Reconstruction view of the scene to compare with earlier MVS methods ?

3. Limited evaluation on outdoor scenes, would be great to see the efficacy of the model on indoor dataset such as ScanNet or NYU.

**Questions:**

Please refer to Weakness Section

---

### Official Review · Reviewer_dNBC · 2024-11-04

**Soundness:** 2
**Presentation:** 3
**Contribution:** 2
**Rating:** 5
**Confidence:** 4

**Summary:**

In this paper, the authors proposed a novel self-supervised multi-frame depth estimation architecture that incorporate the CRAFT module to compress and refine the cost volume through attention mechanism and feature aggregation. The main argument is that training cross-attention layers for image reconstruction facilitates the implicit learning of a warping function, resembling the explicit epipolar warping used in existing methods. Also, by employing masked image modeling for pre-training, the authors can successfully leverage the cross-attention map as a full cost volume for depth prediction in dynamic scenarios without requiring additional information such as camera pose. In experimental section, the authors demonstrated that the proposed method can outperform traditional methods utilizing epipolar-based cost volume in challenging scenarios.

**Strengths:**

First of all, the motivation for proposing a new model to address the limitations of existing epipolar-based methods in handling dynamic objects and the need for additional models is well-founded.

While it may lack significant technical novelty, it shows a solid understanding of the operation of existing modules and effectively applies them to the target problem. More specifically, applying cross-attention mechanism for full cost volume calculation and incorporating MIM to improve matching similarity are highly appropriate choices.

In experimental section, the proposed method achieved SoTA results in depth estimation benchmarks compared with existing self-supervised multi-frame depth prediction methods, demonstrating its effectiveness.

**Weaknesses:**

One concern is the validity of the proposed CRAFT method for better representation learning. While the full cost volume is well-represented through a clear understanding of the cross-attention module, it is not clear whether the proposed module is operated to function as the author’s intention, because all experimental results include MIM, without an independent evaluation of the CRAFT module itself. Furthermore, the significant performance gap between with and without CroCO pre-training raises further doubts about whether the proposed module is functioning as intended.


Another concern is the lack of explanation regarding the justification for cross-attention. I think that the authors should provide a more descriptive explanation as to why Transformer is advantageous for feature matching compared to CNN. Although feature similarity is explicitly calculated through the cross-attention module, the relatively weaker inductive bias of Transformers does not necessarily align with the effectiveness for feature matching.

**Questions:**

I mentioned all comments including reasons and suggestions in the above sections. I recommend that the author will provide all the concerns, and improve the completeness of the paper. If the rebuttal period resolves the above-mentioned concerns, I will gladly raise my score. Also, there are little vague sentences and grammatical errors in the paper. I recommend that the author will revise the paper.

---

### Note · Authors · 2024-11-18

**Comment:**

Thank you for leaving positive reviews. However, we have decided to withdraw our submission. Thank you.

**Withdrawal Confirmation:**

I have read and agree with the venue's withdrawal policy on behalf of myself and my co-authors.